# A multi-model analysis of teleconnected crop yield variability in a range of cropping systems

Matias Heino[1], Joseph H.A. Guillaume[1], Christoph Müller[2], Toshichika Iizumi[3], Matti Kummu[1]

[1]Water and Development Research Group, Aalto University, Finland. Tietotie 1E, 02150 Espoo, Finland

[2]Potsdam Institute for Climate Impact Research, Member of the Leibniz Association, 14473 Potsdam, Germany

[3]Institute for Agro-Environmental Sciences, National Agriculture and Food Research Organization, 3-1-3 Kannondai, Tsukuba, 305-8604 Japan

*Correspondence to*: Matias Heino (matias.heino@aalto.fi), Matti Kummu (matti.kummu@aalto.fi)

**Abstract.** Climate oscillations are periodically fluctuating oceanic and atmospheric phenomena, which are related to variations
in weather patterns and crop yields worldwide. In terms of crop production, the most widespread impacts have been observed for the El Niño Southern Oscillation (ENSO), which has been found to impact crop yields in all continents that produce crops, while two other climate oscillations - the Indian Ocean Dipole (IOD) and the North Atlantic Oscillation (NAO) - have been shown to impact crop production especially in Australia and Europe, respectively. In this study, we analyse the impacts of ENSO, IOD and NAO on the growing conditions of maize, rice, soybean and wheat at the global scale, by utilizing crop yield
data from an ensemble of global gridded crop models simulated for a range of crop management scenarios. Our results show that, while accounting for their potential co-variation, climate oscillations are correlated with simulated crop yield variability to a wide extent (half of all maize and wheat harvested areas for ENSO) and in several important crop producing areas, e.g. in North America (ENSO, wheat), Australia (IOD & ENSO, wheat) and northern South America (ENSO, soybean). Further, our analyses show that higher sensitivity to these oscillations can be observed for rainfed, and fully fertilized scenarios, while the
sensitivity tends to be lower if crops were to be fully irrigated. Since, the development of ENSO, IOD and NAO can potentially be forecasted well in advance, a better understanding about the relationship between crop production and these climate oscillations can improve the resilience of the global food system to climate related shocks.

## 1 Introduction

Climate oscillations are periodically fluctuating oceanic and atmospheric phenomena, and they have been shown to impact hydroclimatological conditions (Dai et al. 1998, Hurrell et al. 2003, Saji and Yamagata 2003, Trenberth 1997, Ummenhofer et al. 2009, Ward et al. 2014) as well as crop productivity (Anderson et al. 2017, Ceglar et al. 2017, Heino et al. 2018, Iizumi et al. 2014, Yuan and Yamagata 2015) worldwide. The most notorious climate oscillation, the El Niño Southern Oscillation (ENSO), is the most significant driver of global climate variability (Trenberth 1997), while two other prominent and widely studied climate oscillations, the Indian Ocean Dipole (IOD) (Saji et al. 1999), and the North Atlantic Oscillation (NAO) (Hurrell 1995), are also known to affect temperature and precipitation patterns around the globe (Hurrell et al. 2003, Saji and Yamagata 2003).

All of these three climate oscillations have been shown to significantly impact crop productivity in global (Heino et al. 2018, Iizumi et al. 2014) as well as regional studies (Anderson et al. 2017, Ceglar et al. 2017, Yuan and Yamagata 2015). The IOD, for example, strongly affects Australia's drought patterns (Ummenhofer et al. 2009) and crop production (e.g. wheat (Yuan and Yamagata 2015)), while NAO has been shown to impact crop productivity particularly in Europe (Ceglar et al. 2017), but also in the Middle East, Northern Africa and some parts of Asia (Heino et al. 2018, Wang and You 2004). However, the largest fingerprint of these three oscillations is that of ENSO, which has been found to influence crop productivity in all continents that produce crops (Anderson et al. 2019, Iizumi et al. 2014).

As the phase and development of ENSO, IOD and NAO can potentially be forecasted from several months (IOD, NAO (Luo et al. 2008, Scaife et al. 2014)) up to one year (ENSO (Luo et al., 2005, Ludescher et al. 2014)) in advance, considerable possibilities arise from understanding the impacts of these climate oscillations on crop production. If these impacts were better understood, it would allow national food agencies, international aid organizations, as well as food industries and farmers to prepare for varying crop development conditions. This would yield great benefits in increasing the resilience of the global food system to climate related shocks.

Until now, global scale studies about the relationship between crop production and climate oscillations have relied on individual satellite-based (Iizumi et al. 2014) or single-model simulated (Heino et al. 2018) crop yield estimates. The data produced and published in phase 1 of the global gridded crop model intercomparison (GGCMI) of the Agricultural Model Intercomparison and Improvement Project (AgMIP) now allows conducting assessments related to crop yield variability with an ensemble of models and a range of fertilizer use and irrigation set-ups (Elliott et al. 2015, Müller et al. 2019). Given the large variation in crop yield estimates across models (Müller et al. 2017, Rosenzweig et al. 2014), using an ensemble of models can allow for more robust estimates with better quantification of uncertainty in estimated yield impacts than using a single model.

By using the historical crop yield output derived from a multi-model ensemble of GGCMI, we aim to analyse the impacts of ENSO, NAO and IOD on maize, rice, soybean and wheat yields at the global scale. This extends previous studies, which are based on crop yield estimates from single datasets (Heino et al. 2018, Iizumi et al. 2014) and have assessed the impacts of solely ENSO (Iizumi et al. 2014) or the impacts of multiple oscillations on an aggregated crop productivity proxy (Heino et al. 2018). Further, since it is well known that agricultural management can have a major influence on climate induced crop yield variations (Challinor et al. 2014, Müller et al. 2018a), we assess these impacts in different irrigation and fertilizer use scenarios. As a result, we are able to highlight potential management options to mitigate the impacts of these oscillations on crop production. In the Results and Interpretation section we also compare our results with previous work in order to provide a comprehensive overview of known phenomena while avoiding repetition.

## 2 Data and methods

### 2.1 Physically simulated crop yield data

Global data of physically simulated maize, rice, soybean and wheat yield (t ha$^{-1}$) were obtained from the global gridded crop models' (GGCMs) simulations included in phase 1 of the GGCMI of AgMIP (Elliott et al. 2015, Müller et al. 2019). While most of the 12 models included here simulate the growth of all four target crops, a few simulate only some (Table 1): EPIC-TAMU (maize and wheat), pAPSIM (maize, soybean and wheat), and PEGASUS (maize, soybean and wheat). A recent study evaluated the performance of the models, included in the GGCMI of AgMIP, in reproducing reported historical yield anomalies, and did not find any GGCM clearly superior to any other (Müller et al. 2017, Figure S1), thus highlighting the benefits of utilizing a model ensemble in yield variability assessments to account for uncertainty in individual model results.

**Table 1. Crop yield data used in this study. 'All' refers to all of the crops included in this study, i.e. maize (M), rice (R), soybean (S) and wheat (W). Three model configurations were utilized: harmonized growing season and nutrient input (fullharm), harmonized growing season and no nutrient limitation (harm-suffN), and standard model-specific assumptions (default). Details about the climate forcing data availability are given in the footnotes.**

| | Crops included for different model configurations | | | | |
| --- | --- | --- | --- | --- | --- |
| | fullharm | harm-suffN | default | Model Reference | Data reference |
| **CGMS-WOFOST** | - | - | All[1] | (de Wit and Van Diepen 2008) | Hoek and de Wit (2018a, b, c, d) |
| **EPIC-Boku** | All[1,2] | All[1,2] | All[1,2] | (Izaurralde et al. 2006, Williams 1995) | Schmid (2018a, b, c, d) |
| **EPIC-IIASA** | All[1] | All[1] | All[1] | (Izaurralde et al. 2006, Williams 1995) | Balkovic et al. (2018a, b, c, d) |
| **EPIC-TAMU** | M, W[1,2] | M, W[1,2] | - | (Izaurralde et al. 2012) | Reddy et al. (2018a, b) |
| **GEPIC** | All[1] | All[1] | All[1] | (Folberth et al. 2012, Liu, J. et al. 2007, Williams 1995) | Folberth (2018a, b, c, d) |
| **LPJ-GUESS** | - | All[1,2] | All[1] | (Lindeskog et al. 2013, Smith et al. 2001) | Pugh et al. (2018a, b, c, d) |
| **LPJmL** | - | All[1,2] | All[1,2] | (Bondeau et al. 2007, Waha et al. 2012) | Müller (2018b, c, d, e) |
| **ORCHIDEE-crop** | M[1,3], R[1,3], S[1,3], W[1] | M[1,3], R[1], S[3], W[1] | M[1], R[1,3], S[1], W[1] | (Wu et al. 2016) | Wang and Ciais (2018a, b, c, d) |
| **pAPSIM** | M, S, W[1,2] | M, S, W[1,2] | M, S, W[1,2] | (Elliott et al. 2014, Keating et al. 2003) | Elliott (2018a, b, c) |
| **pDSSAT** | All[1,2] | All[1,2] | All[1,2] | (Elliott et al. 2014, Jones et al. 2003) | Elliott (2018d, e, f, g) |
| **PEGASUS** | M, S, W[1,2] | M, S, W[1] | M, S, W[1] | (Deryng et al. 2011, Deryng et al. 2014) | Deryng (2018a, b, c) |
| **PEPIC** | All[1] | All[1] | All[1] | (Liu, W. et al. 2016, Williams 1995) | Liu and Yang (2018a, b, c, d) |

1) AgMERRA, Timespan: 1980-2010

2) Princeton, Timespan: 1948-2008

3) Princeton, Timespan: 1979-2010

5    Yield variability in the GGCMs included in GGCMI is mainly driven by weather circumstances and $CO_2$ concentration, while soil conditions and agricultural management practices are considered static (Müller et al. 2019). To account for varying assumptions of growing season and fertilizer use, in GGCMI, model simulations were conducted for three configurations:

standard model assumptions (default), harmonized growing season and nutrient input (fullharm), and harmonized growing season with no nutrient limitation (harm-suffN). For the default configuration each modelling group used their own model assumptions. In the harmonized model set-ups, crop planting and harvesting dates were standardized among the models and are literature-based (Elliott et al. 2015), while fertilizer application rates are either unlimited (harm-suffN) or based on

published data (fullharm). Further, all of the GGCMI simulation results are provided separately for irrigated and rainfed conditions. In the irrigated simulation settings, no restrictions on water availability are considered (Müller et al. 2019). In GGCMI, the models simulate only a single growing season per year. Two models included in the GGCMI archive, PRYSBI2 and CLM-Crop, were excluded from this study because either the harmonization of growing season provided unreliable results (CLM-Crop) or the model does not distinguish between rainfed and irrigated crops (PRYSBI2).

The "actual" cropping scenario, used in the main analyses (with literature based shares of rainfed and irrigated areas, see Sect. 2.3), utilizes the fullharm set-up, and the harm-suffN setting for LPJ-GUESS and LPJmL, which do not consider nitrogen limitation and thus cannot harmonize on fertilizer settings (Table 2). For comparison, the sensitivity analysis (see Sect 2.4) for the actual cropping scenario was repeated with the default model set-up (see Supplement), while the harm-suffN scenario was used in assessing the impacts of the oscillations in fully fertilized conditions.

**Table 2. The management scenarios used in this study. The actual set-up is used in the main analyses, while the fully irrigated, rainfed, fully fertilized, and the fully irrigated and fertilized management scenarios are used for comparing the impacts in different cropping systems.**

| Management scenario | Irrigated areas | Fertilizer use |
|---|---|---|
| **Actual** | Literature based | Literature based* |
| **Fully irrigated** | All areas irrigated | Literature based |
| **Rainfed** | No areas irrigated, all areas rainfed | Literature based |
| **Fully fertilized** | Literature based | Fully fertilized |
| **Fully irrigated and fertilized** | All areas irrigated | Fully fertilized |

**\*)** For LPJ-GUESS and LPJmL, limitations on fertilizer use are not considered. These models are excluded from the "Actual" scenario for the comparison with varying fertilizer use.

This study utilizes simulations driven with two historical meteorological forcing data sets (bias-corrected re-analysis weather data sets): AgMERRA (Ruane et al. 2015) and Princeton Global Forcing data set (Sheffield et al. 2006) (Table 1). AgMERRA

was selected as the main climate input for this study, as a large number of GGCMs supplied data for this climate forcing data set, while the Princeton data was selected for reference due to its long timespan and previous use in a similar study (Heino et al. 2018). A detailed description of the GGCMI phase 1 modelling protocol can be found in Elliott et al. (2015) and the output data set is described by Müller et al. (2019).

**2.2 Climate oscillation data**

To represent the historical fluctuations of ENSO, IOD and NAO, the following indices were chosen: the Japan Meteorological Agency (JMA) SST Index (Florida State University 2015), the SST Dipole Mode Index (NOAA Earth System Research Laboratory 2017, Saji et al. 1999), and Hurrell's North Atlantic Oscillation Index (primary component (PC)-based) (Hurrell 1995, National Center for Atmospheric Research 2015), respectively. These indices were selected because they are all well established and have already been used in several studies related to crop production (Heino et al. 2018, Kim and McCarl 2005, Yuan and Yamagata 2015). For ENSO, the Niño 3.4 index (NOAA Earth System Research Laboratory 2019) was also tested given its common use in ENSO related studies (Stuecker et al. 2017, Zhang et al. 2015), with results shown in the Supplement. The indices were transformed to annual values by calculating the mean index for the months when the oscillations tend to have the strongest signal, according to existing sources, (i.e. December (year t), January (t+1), February (t+1) for ENSO (Trenberth 1997) and NAO (Hurrell et al. 2003); September, October and November (year t for all) for IOD (Saji et al. 1999). This therefore only tests for relationships with a phase-locked measurement of the oscillation rather than investigating intra-annual temporal effects. Using seasonal or monthly data increases the number of significance tests for a given location and therefore the risk of false positives, and interpretation of results would require understanding of how climate oscillations, local weather conditions and yield are connected over time. However, it requires accurate, high-resolution global crop calendars which are not available. Finally, in order to make the oscillation indices comparable with each other, each oscillation index time series was standardized (by subtracting the average index value from the annual values and dividing by their standard deviation).

**2.3 Crop yield data aggregation and de-trending**

The gridded crop yields were allocated to annual yields based on the sowing dates used in the harmonized GGCMI simulations. The "harvest year t" is assigned to crop yields which are sown between May of the actual year (t) and April of the next year (t+1). This definition for harvest years was selected, because it ensures that the average lifespans of all these oscillations are within the harvest year, and thus many of the major known teleconnections of these oscillations during the crop growing season are included in the analysis (e.g. in Australia, Africa, and South America).

The crop yield data were aggregated spatially to the geographical scale of Food Production Units (FPUs), which divide the world into 573 spatial units that are hybrids of river basins and administrative (economic) areas (Kummu et al. 2010). For the actual cropping scenario, rainfed and irrigated crop yields were combined by calculating the mean yield as the total production divided by the total harvested area across both cropping systems, using literature based values about harvested area (Portmann et al. 2010). The aggregation for fully irrigated and rainfed scenarios was conducted similarly by dividing total production by harvested areas but assuming that all cropland is either irrigated or rainfed, respectively.

In order to extract the interannual variability of the crop yield data, they were de-trended. This was conducted by subtracting a five-year moving average yield from the annual yield values (three-year average at both ends of the time series), similarly to

several previously conducted studies about yield variability (Iizumi et al. 2014, Iizumi and Ramankutty 2016, Müller et al. 2017, Müller et al. 2018a). The anomalies were then divided by five-year (or three-year) averages to obtain proportional annual deviation from the normal values. The equation of the procedure is shown below:

$$\Delta Y_{f,s,m,c,t} = \frac{Y_{f,s,m,c,t} - \bar{Y}_{f,s,m,c,t}}{\bar{Y}_{f,s,m,c,t}} * 100, (1)$$

where $\Delta Y_{f,s,m,c,t}$ denotes relative yield anomaly for each FPU ($f$), scenario ($s$), model ($m$), crop ($c$) and year ($t$) compared to the average yield ($\bar{Y}_{f,s,m,c,t}$) for the moving time window around year t. The use of a shorter time window at the beginning and end of the yield time series allows longer de-trended time series, and it is assumed that it would rarely lead to errors about the sign of yield anomalies and thus the derived relationships between climate oscillation and yield anomalies. Other studies have tested other de-trending methods as well, but have found no major impact from the method selected (Iizumi et al. 2014, Iizumi

and Ramankutty 2016, Müller et al. 2017).

## 2.4 Crop yield sensitivity to the oscillations

The sensitivity of actual crop yield to the oscillations was investigated using a multivariate linear regularized ridge regression model, with the oscillation indices as explanatory variables and the annual crop yield anomalies as dependent variable. The ridge regression framework was selected because it allows accounting for correlations among the explanatory variables (here

oscillation indices). For the main analysis (actual scenario), the regression was calculated for each FPU separately using crop yield anomaly time series from all GGCMs that simulate the crop in question with the AgMERRA climate input (N=216-297, depending on crop). Hence, we utilize the crop yield time series of all the models in fitting the regression. In the regression model, the slope coefficients represent sensitivity. The optimal regularization value, for the regression, was selected by performing a generalized cross-validation (tested regularisation values ranged between $10^{-6}$ and 10).

The existence of significant relationships was assessed by calculating a multivariate ridge regression from random bootstrap samples (N = 1,000, with replacement) of crop yield–oscillation index combinations. Statistical significance therefore tests the robustness of observed ridge regression coefficients across different samples drawn from the time series. The optimal regularization value was selected for each bootstrap sample as described above, which follows the principle described in Abram et al. (2016). The linear relationship was defined to be significant ($p < 0.1$), if 95 % (two-sided test) of the sampled sensitivity

values were either larger or smaller than zero. Thus, a 10 % probability was accepted of wrongly classifying a linear relationship as significant. Note that the relatively high risk level in statistical regression ($p < 0.1$) is commonly used in global climate-yield analysis because of the limited access to high quality yield data at the global scale (e.g., Ray et al., 2015). To check robustness of results, the same analysis was also conducted utilizing the crop yield data derived using the Princeton climate input, different model configurations as well as individual models and average weather (soil moisture and temperature)

conditions (Martens et al. 2017, Ruane et al. 2015) during the growing season. Further, to illustrate the effect of using phase-locked indices rather than investigating intra-annual temporal variation (see Section 2.2), the sensitivity of crop yield to these oscillations was also assessed by using the average harvest season oscillation indices as explanatory variable (see Supplement).

## 2.5 Average crop yield anomalies during strong oscillation phases

The crop-specific average yield anomalies observed during strong oscillation phases were investigated for the actual cropping scenario. The crop yield changes that occur during years when the oscillations are in their strong phases were summarized by the median crop yield anomaly (in percent) of those years. The median anomaly was calculated using all the GGCMs that simulate the crop in question (N=216-297, depending on crop) for the actual scenario with AgMERRA climate input. Strongly negative (positive) phases of the oscillations were defined as the years when the respective oscillation index was smaller (larger) than the 25th (75th) percentile of all yearly index values ($N_{anomaly}$ = 54-74, depending on crop). The statistical significance ($p < 0.1$) of the changes was assessed by bootstrapping (n = 1,000, with replacement) the crop yield anomalies, and calculating the median of each bootstrap sample. If over 95 % (two-tailed test) of the sample of medians were either larger or smaller than zero, the change was considered statistically significant. Statistical significance therefore tests the robustness of observed anomalies across different samples drawn from the time series.

## 2.6 Impacts in different cropping systems

To assess how expanding or reducing the extent of irrigated area, and increasing fertilizer use would change the impacts of climate oscillations on crop yields, compared to the actual scenario, the main sensitivity analysis (see description above – Sect. 2.4) was conducted for a set of scenarios (Table 2): i) all cropland was only rainfed (with fullharm setup), ii) all cropland was fully irrigated (fullharm), iii) all cropland was fully fertilized (actual irrigation with fullharm-suffN), and iv) all cropland was fully irrigated and fertilized (fully irrigated with harm-suffN). In addition to analysing how the above mentioned four scenarios compare against the actual scenario, the fully irrigated and rainfed scenarios were also compared. To quantify how the impacts in these cropping systems vary, average sensitivity magnitudes were compared for each crop. Specifically, for a pair of scenarios, the average difference of their absolute sensitivity values was calculated across all oscillations and FPUs, where at least one of the scenarios shows a significant sensitivity. To obtain a measure relative to the actual (or irrigated when comparing irrigated and rainfed scenarios) scenario, the average difference values were divided with the average sensitivity magnitude of the actual (or irrigated) scenario for the FPUs included. The corresponding equation is:

$$\Delta S_{s12,c} = \sum_{o,f} \left. \frac{|S_{s1,c,f,o}| - |S_{s2,c,f,o}|}{|S_{s1,c,f,o}|} \right/ n_{F,O} * 100 \%, (2)$$

where at least of one $|S_{f,s1,c,o}|$ or $|S_{f,s2,c,o}|$ is statistically significant. $f$, $s \in \{s1, s2\}$, $c$, and $o$ are indices of FPU, management scenario, crop, and oscillation respectively. $\Delta S_{s12,c}$ denotes the average proportional sensitivity difference of each crop ($c$)

between the scenarios, while $S_{f,s1,c,o}$ and $S_{f,s2,c,o}$ represent the sensitivity in the respective management scenarios $s1$ and $s2$. $n_{F,O}$ is the number of cases (oscillation and FPU) where at least one of the scenarios has a significant sensitivity.

For each crop, to assess whether the mean sensitivity magnitude difference is statistically significantly different from zero, a
distribution of the mean sensitivity magnitude difference was created by calculating the average from bootstrapped (N = 1,000, with replacement) difference values of each FPU and oscillation. For the comparisons with varying fertilizer use, only those nine GGCMs which have data for both 'fullharm' and 'harm-suffN' settings and thus simulate nutrient stress (Table 1), were included.

## 3 Results and Interpretation

### 3.1 Global extent of climate oscillation impacts

Globally, climate oscillations have widespread effects on crop yields (Table 3), but both the direction and magnitude of impacts vary spatially and across crops (Fig. 1). Out of the oscillations studied here, ENSO shows the widest impacts on yields of maize (statistically significant sensitivity in 51 % of harvested areas), wheat (49 %) and rice (48 %), while IOD and ENSO both show a similar extent of impacts on the yields of soy (53 % and 50 %, respectively) (Table 3). Generally, NAO seems to
have the smallest impacts on the yields of the crop types inspected here in terms of harvested areas, although it still shows relatively strong influence on wheat (42 %) and maize (35 %) yields. In terms of sensitivity direction, it is notably more widespread for yield to increase towards the positive phase of ENSO (i.e. El Niño) for all crop types inspected here (i.e. positive sensitivity). For IOD and NAO, the results are more mixed, though both show larger harvested areas where yield decreases towards the positive phase for maize. These results align with crop yield anomalies during strong oscillation phases, as they
also show widespread average impacts (Table S2).

**Table 3. Extent of significant sensitivity. Crop-specific harvested area (10⁶ ha) extent (and percent of total crop-specific harvested area), where actual crop yield shows statistically significant positive (+) or negative (-) sensitivity to ENSO, IOD and NAO, i.e. there is a statistically significant (two-sided p-value < 0.1) linear relationship between crop yield anomalies and the studied oscillations (see Methods). Harvested area extent showing significant anomalies is shown in Table S2.**

| Sensitivity | ENSO | | IOD | | NAO | |
|---|---|---|---|---|---|---|
| | − | + | − | + | − | + |
| **Maize** | 30 (20 %) | 46 (31 %) | 36 (24 %) | 18 (12 %) | 44 (29 %) | 10 (6 %) |
| **Rice** | 31 (19 %) | 47 (29 %) | 22 (13 %) | 16 (10 %) | 1 (0 %) | 8 (5 %) |
| **Soybean** | 6 (8 %) | 31 (42 %) | 32 (42 %) | 8 (11 %) | 22 (29 %) | 6 (8 %) |
| **Wheat** | 28 (13 %) | 77 (36 %) | 45 (21 %) | 46 (21 %) | 20 (10 %) | 69 (32 %) |

## 3.2 Impacts in different areas

ENSO's relationship with crop yield seems to provide the most distinct spatial patterns across the crop types, crop models and oscillations studied here (Figs 1-3, Figs S4-S10, Supplementary zip-file). Crop yields tend to decrease towards the positive phase of ENSO (El Niño) in a large proportion of sub-Saharan Africa, as well as eastern parts of South America and Australia while yields seem to increase towards the positive phase on the coast of Peru and North America (Fig. 1, global regions mapped in Fig. S1). In general, these results align well with the spatial patterns found in existing studies on the Palmer Drought Severity Index (Dai et al. 1998) as well as soil moisture and temperature anomalies (Figs S2-S3). Also, in terms of model and methodological agreement, consistent increase (decrease) of wheat yields in parts of the Middle East can be observed for ENSO towards its positive (negative) phase (Figs 1-3, Figs S11-S12). For the southern tip of Africa, wheat and soybean seem to be related with opposite impacts. This is potentially because of differences in harvest timing and the related weather conditions; wheat is harvested in the autumn, while soybean is harvested the following spring, and thus is more exposed to the drier conditions related to ENSO during the boreal winter (Fig. S2, Philippon et al. 2012).

When comparing our results to a study about ENSO's crop yield impacts, which utilized satellite-based crop yields (Iizumi et al. 2014, Iizumi et al. 2018a), the agreement of the impacts varies. Our results agree with existing studies for example for large parts of Africa and eastern Asia, where El Niño is mostly related to negative impacts, while results do not agree in North America (wheat, maize) and Australia (maize). However, it should be noted, that these differences are no surprise, since it has been shown that only a third of global crop yield variability can be attributed to seasonal climate variation (Ray et al. 2015). In contrast to the satellite-based data, the models used here deliberately focus on weather impacts on crop yield, and do not consider the impacts of e.g. multiple cropping or weather triggered pest outbreaks and management responses, which can also be major contributors to crop yield variations.

For IOD, strong and consistent impacts (Figs 1-2) among crop models (Fig. 3) can be observed in eastern Australia, especially for soybean and wheat (Figs 1-3), where the IOD is related to drier and warmer weather conditions (Figs S2-S3). This corroborates a previous study conducted on the relationship between IOD and wheat yields, which showed that around 40 % of Australia's wheat yield variability can be attributed to the IOD (Yuan and Yamagata 2015), and where the oscillations are

together also able to explain a substantial portion of crop yield variability (>25 %, Fig. S13). Further, consistent results among models and methods (Figs 1-3) for IOD can be observed in parts of Eastern Europe and Central Asia, where the positive (negative) phase of the IOD is related to an increase (decrease) in wheat, maize and soybean yields. In Southeast Asia, and southern Africa, the impacts of the IOD vary between crops. For example, in Southeast Asia, rice shows a positive sensitivity (increasing yield towards the positive phase) while maize and soybean show a negative one. In eastern China, maize, wheat

and soybean yield variability seems to be related to the IOD to some extent. However, these relationships are less certain, as they are not consistently found by the majority of the individual models (Fig. 3).

For NAO, the relationships are generally less certain in terms of model agreement compared to ENSO and IOD (Fig. 3). NAO's most significant impacts can be observed in eastern Europe and the Middle East for maize, soybean and wheat yields (Figs 1-2). In the Middle East, the sensitivity of wheat and maize (soybean) yield to NAO seems to be negative (positive), while mostly

positive sensitivity is found in Europe and western Russia for maize, soybean and wheat. These differences between crop types observed in the Middle East can potentially be due to differing growing seasons. In the models, the sowing dates of soybean vary strongly in space and between irrigation regimes (for some areas soybean sowing occurs in Spring before May, while in other areas soybean is planted later in the year), which can have an effect on the observed signal compared to maize and wheat. In general, the patterns observed in eastern Europe, western Russia and the Middle East align well with results from previous

studies about crop productivity and weather variations (Cullen et al. 2002, Heino et al. 2018, Hurrell et al. 2003). Although, the results for NAO are relatively similar between different model configurations (Figs S4-S6), the results are not as consistent among the GGCMs as for the other oscillations (Fig. 3).

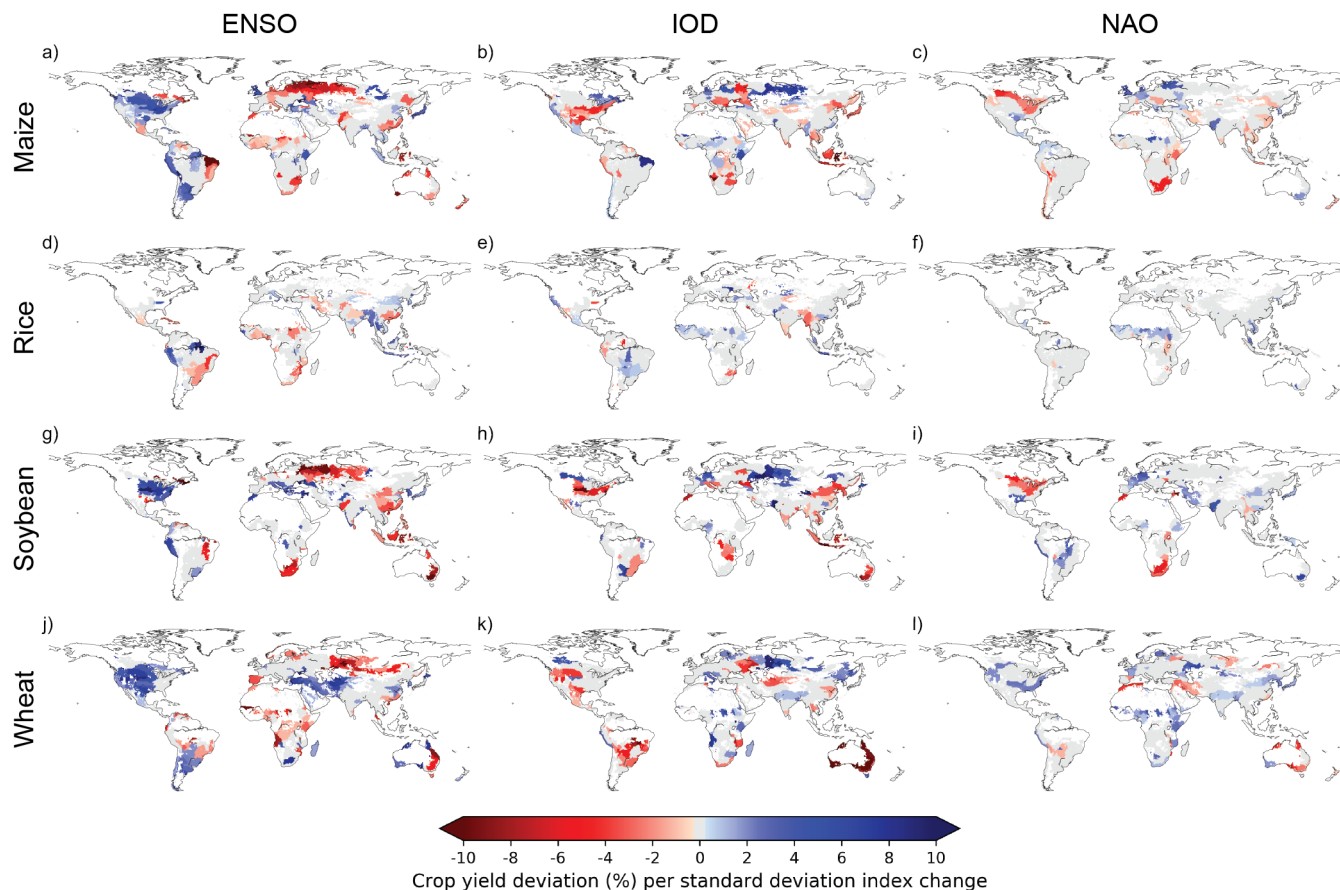

**Figure 1. Actual maize (a-c), rice (d-f), soybean (g-i), and wheat (j-l) yield sensitivity to ENSO, IOD and NAO at FPU scale. The sensitivity values are derived using crop yield data from all GGCMs that simulate the crop in question with the AgMERRA climate input. Statistically insignificant (p > 0.1) sensitivity values are marked as zero (colour grey). White colour denotes that the crop in question is not produced in that area. Results with Princeton climate input and default model setup are shown in Figs S4 and S5, respectively. Median, maximum and minimum sensitivities as well as consistency across individual models are shown in Figs S7-S9, respectively. Results for individual models are shown in a Supplementary zip-file. Results with oscillation indices calculated in the harvest season are shown in Fig. S11, with the associated seasons shown in Fig. S12.**

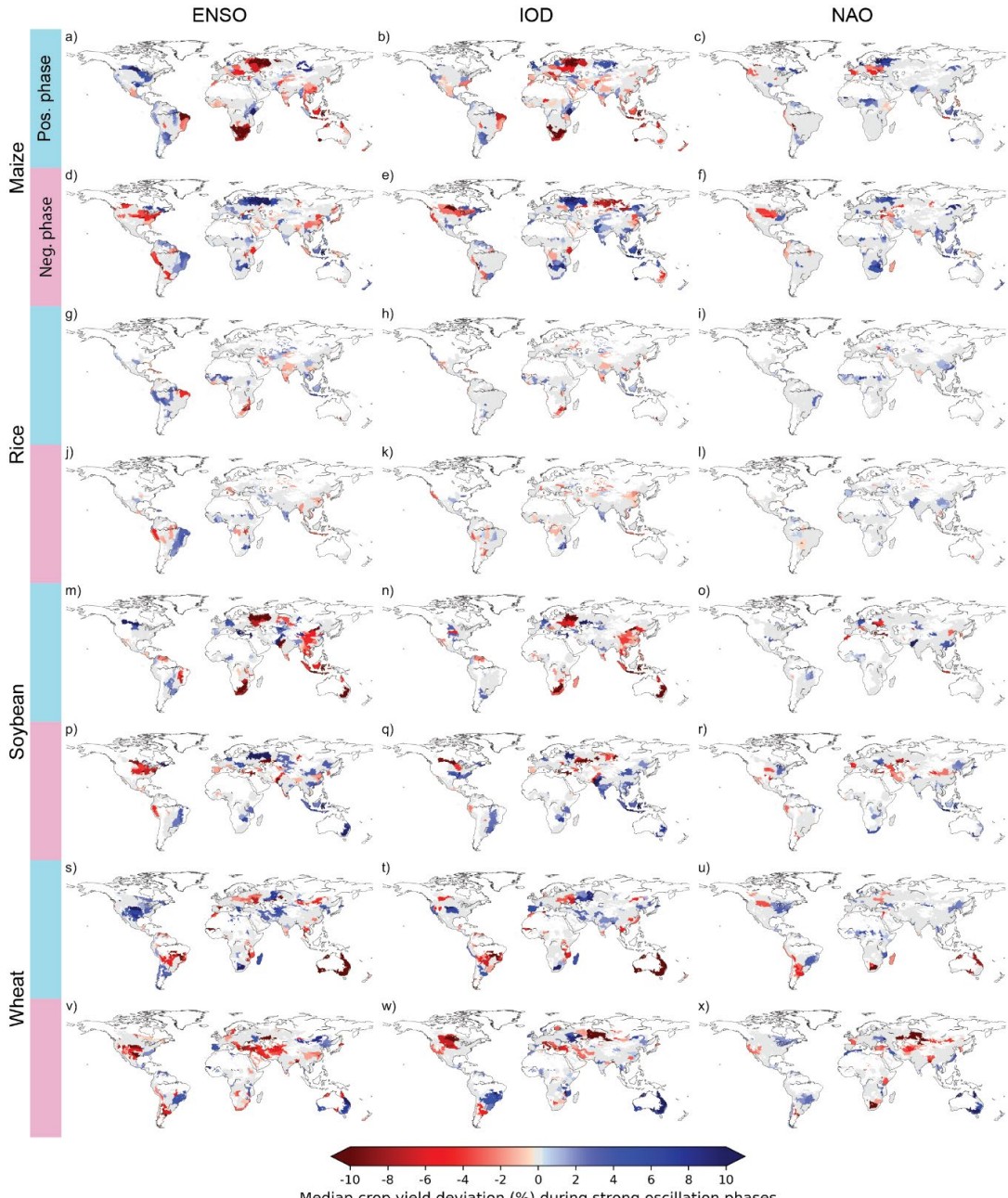

**Figure 2. Actual maize (a-f), rice (g-l), soybean (m-r), and wheat (s-x) yield anomalies during strong phases of ENSO, IOD and NAO at FPU scale. The anomaly values are derived from a sample including crop yield data from all GGCMs that simulate the crop in question with the AgMERRA climate input. Statistically insignificant (p > 0.1) anomaly values are marked as zero (colour grey). White colour denotes that the crop in question is not produced in that area. Patterns are discussed in Sections 3.1 and 3.2.**

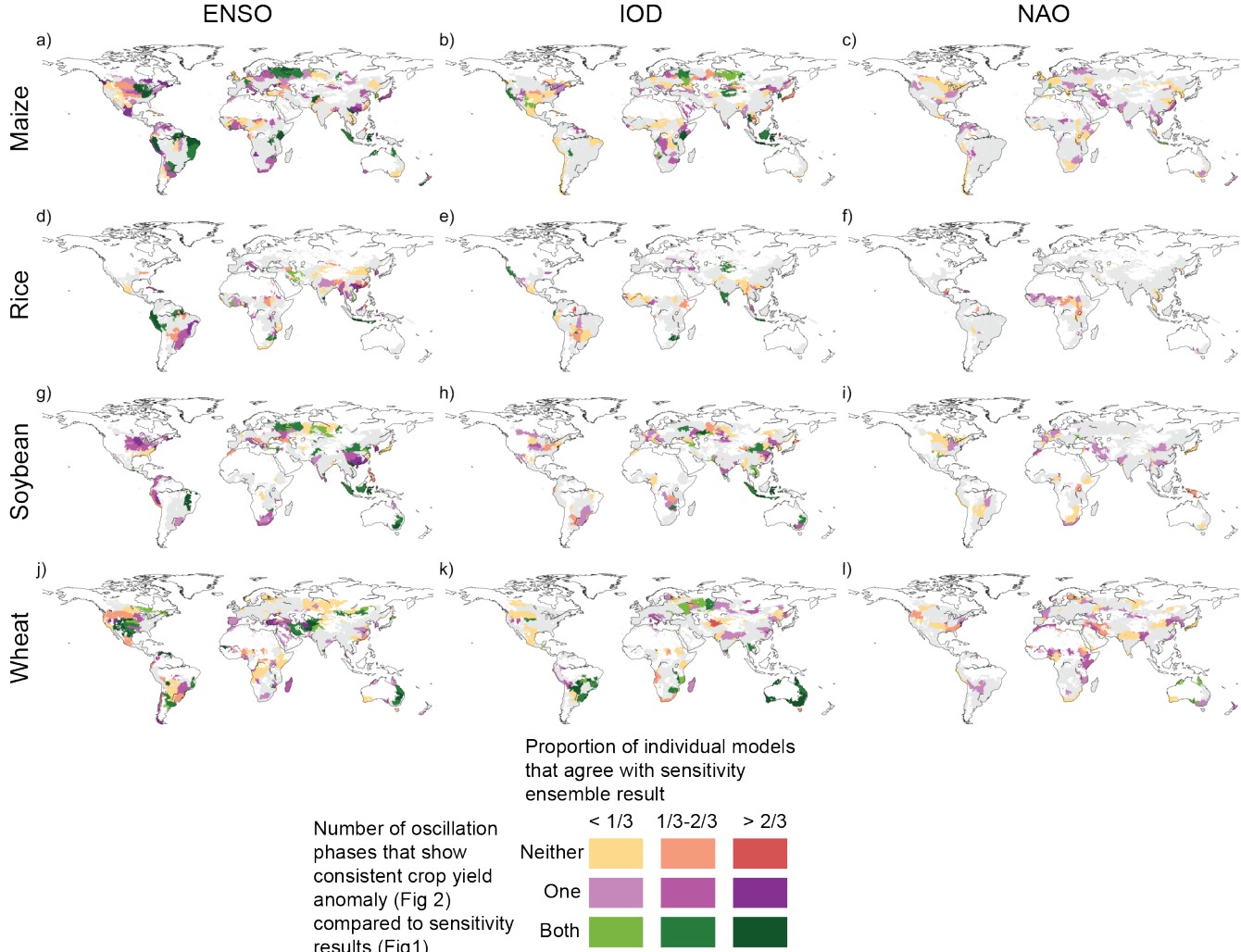

**Figure 3. Summary of relationship between ENSO, IOD, and NAO and crop yields across models and methods for maize (a-c), rice (d-f), soybean (g-i), and wheat (j-l). The y-axis of the colour bar shows whether there is agreement between the sensitivity analysis (Fig. 1) and the average anomalies during strong oscillation phases (Fig. 2): '*Neither*' denotes that the strong oscillation phases are not related to significant average crop yield anomalies that are consistent with the sensitivity analysis, '*One*' means that either positive or negative oscillation phase shows a significant average anomaly that is consistent with the sensitivity result (e.g. positive sensitivity, and positive anomaly during a positive oscillation phase), '*Both*' means that both phases of the oscillations show consistent average anomalies during the strong oscillation phases (e.g. positive anomaly during a positive oscillation phase, and negative anomaly during a negative oscillation phase in an FPU with positive sensitivity). The x-axis of the colour bar shows the proportion of individual models that show significant sensitivity of same sign compared to the result from the ensemble analysis (see Fig. 1 above, Fig. S10 and Supplementary zip-files). Areas where the ensemble results do not show a statistically significant relationship are marked in grey, while white colour denotes that the crop in question is not grown in that area**

### 3.3 Magnitude of impacts in different cropping systems

Irrigation plays a key role in reducing crop yield sensitivity to climate oscillations, with yield varying up to three times more (for wheat) across the range of oscillations when comparing fully irrigated and rainfed scenarios (Fig. 4). Comparing rainfed to actual conditions shows that irrigation has already substantially reduced the effects of climate oscillations on crop yields. The average difference in sensitivity is largest for rice, where average sensitivity would be over two times higher, i.e. yield would vary two times more across the range of the oscillations if all cropland was rainfed (Fig. 4g). The difference in sensitivity is smallest for soybean (29 %, Fig. 4l), while maize and wheat show a relative increase in sensitivity of 47 % (Fig. 4b) and 60 % (Fig. 4q), respectively. This ranking is expected, as the majority of rice harvested areas are irrigated (62 % globally) and soybean has the smallest irrigated area share of these four crops (8 %), while maize (21 %) and wheat (31 %) fall in between (Portmann et al. 2010).

Conversely, average sensitivity would be reduced if crops were fully irrigated without any limitations on water availability, compared to the actual situation, for all the inspected crop types. Benefits of further irrigation are limited by its current use, which might be why rice shows the smallest difference in average impacts (most of the rice harvested area is already irrigated, Fig. 4h). The average decrease in crop yield sensitivity to the oscillations is largest for wheat (54 %, i.e. yield varying 54 % less across the oscillations compared to actual conditions, Fig. 4r) and soybean (39 %, Fig. 4m), while maize shows a 35 % (Fig. 4c) average decrease.

Unlimited fertilizer (fully fertilized scenario) use yields statistically significantly larger average sensitivity compared to actual conditions for maize (21 %, Fig. 4d), rice (11 %, Fig. 4i) and wheat (18 %, Fig. 4s). For these crops, these climate oscillations have a stronger impact on yields in cropping systems that do not have limitations related to nutrient availability. This reflects previous research that has found increased crop yield variability under additional fertilizer inputs (Müller et al. 2018a). This is potentially because in low crop yield years, fertilizer use is not the main limiting factor, so yields are not significantly improved, while in years when climate conditions are suitable for crop growth, yields become even higher, which would increase the sensitivity value as well (Fig. S24). Note that this does not mean fertilizers fails to improve crop yields – only that it does not lead to more stable yields in the face of weather variability. Soybean has very little change in sensitivity under full fertilisation (Fig. 4n). This is likely because it is a legume and has lower nitrogen requirements (nitrogen availability is not even considered in soybean simulations in some models).

Combining both unlimited irrigation and fertilizers, all of the crop types show smaller average sensitivity compared to the actual cropping system scenario (Fig. 4). The decreased sensitivity due to increased irrigation dominates the increased sensitivity due to increased fertilizer use. However, the differences in sensitivity magnitude are large between crops, with wheat having the largest decreases in average sensitivity magnitude (39 %, Fig. 4t), and rice having the smallest (16 %, Fig. 4j).

The above mentioned results can be observed spatially Supplementary Figs S14-S23), which e.g. clearly show that, in most areas, the sensitivity magnitude is larger in the rainfed as well as fully fertilized scenarios, i.e. yields vary more across the range of the oscillation index. The spatial results also highlight areas with potential to reduce impact of oscillations, for example for ENSO in northern South America (soybean and rice) as well as for IOD in Australia (wheat), where high sensitivity to the respective oscillations can be observed for the actual scenario.

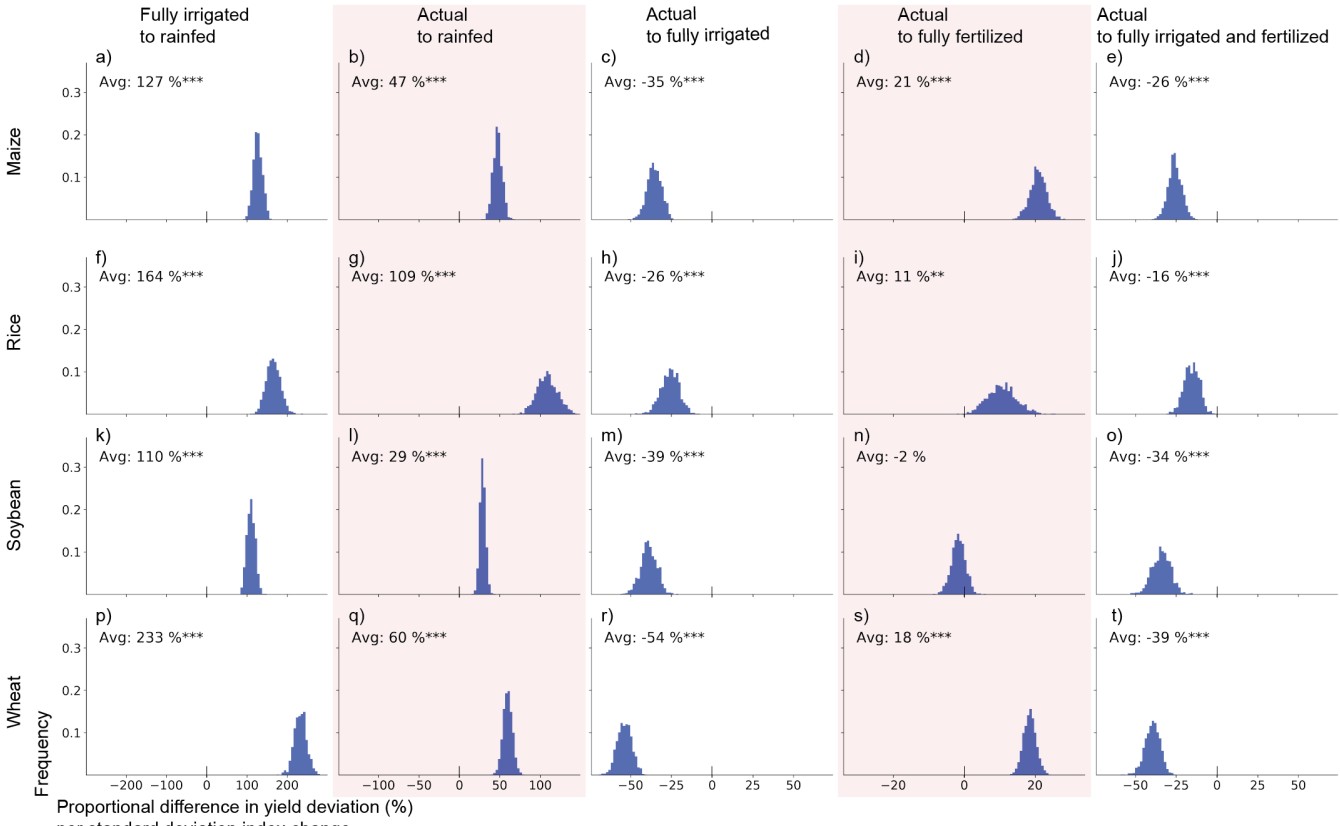

**Figure 4. Relative average difference in sensitivity magnitude of maize (a-e), rice (f-j), soybean (k-o) and wheat (p-t) between a range of cropping scenarios through all the studied oscillations and FPUs. To quantify how the impacts in these cropping systems vary, average sensitivity magnitudes were compared for each crop. Specifically, for a pair of scenarios, the average difference of their absolute sensitivity values were calculated across all oscillations and FPUs, where at least either scenario shows a significant sensitivity. To obtain a measure relative to the actual (or irrigated when comparing irrigated and rainfed scenarios) scenario, the average difference values were divided with the average sensitivity magnitude of the actual (irrigated) scenario for the FPUs included. For each crop, to assess whether the mean sensitivity magnitude difference is statistically significantly different from zero, a distribution of the mean difference was created by calculating the average from bootstrapped (N = 1,000, with replacement) difference values of each FPU and oscillation. For the scenarios with varying fertilizer use set-up, we included only those nine GGCMs which have data for both 'fullharm' and 'harm-suffN' settings and also simulate nutrient stress, i.e. pDSSAT, EPIC-Boku, EPIC-IIASA, GEPIC, pAPSIM, PEGASUS, EPIC-TAMU, ORCHIDEE-crop, and PEPIC. Triple, double and single asterisks denote the confidence level at 99.9 %, 99 % and 90 %, respectively. Maps of sensitivity for each cropping system are shown in Figs S14-S18 and difference in sensitivity magnitude in Figs S19-S23. Please note different scale in x-axis between the columns.**

## 4 Discussion

In this study, we inspected the historical relationship between crop yield variability and climate oscillations in a range of cropping systems by utilizing an ensemble of historical crop yield simulations generated in GGCMI. The results of this study highlight the widespread impacts that ENSO, IOD and NAO have on crop yields at the global scale, as well as potential options for mitigating their impacts. Further, we find robust impacts for these oscillations in many areas around the globe, e.g. in North and South America for ENSO, and in eastern Australia for IOD, where these insights can potentially be utilized in efforts mitigating weather driven variations in crop productivity (Iizumi et al. 2018b).

The reliability and usefulness of these results vary significantly between regions, crops and oscillations. In general, the teleconnections related to ENSO are the strongest, which is sensible, since ENSO has been shown to be the most significant driver of global climate variability (Dai et al. 1998, Trenberth 1997). Various institutions (including the United Nations) already provide action plans to mitigate ENSO's impacts on society. In Australia, there is significant potential to utilize the information of IOD along with ENSO, to understand crop yield fluctuations, as they can explain a large proportion of local crop yield variability (Fig. S13, Yuan and Yamagata 2015). Some promise also exists in using oscillation forecasts in predicting crop yield variability (Nobre et al. 2019). However, the quality of predictions of this type would naturally depend on the skill of the climate forecasts as well as the strength of the teleconnection. This study only provides a first assessment of correlations, and further work is needed before reliable forecasts can be provided.

Our results join existing research (Müller et al. 2018a, Schauberger et al. 2016, Okada et al., 2018) in highlighting the major role of irrigation in mitigating climate related crop yield variations, and thus securing global food production. This is an important point, since water supplies are highly stressed in many important crop-producing areas (Kummu et al. 2016), which are also impacted by climate oscillations, such as parts of North America and South Asia. Thus, diminishing water resources could pose a major barrier in mitigating future negative impacts related to climate oscillations and climate variability in general. This can be very problematic, given that climate change will likely increase the occurrence of extreme weather in the future (Coumou and Robinson 2013). With given water shortages in some regions (Heinke et al. 2019), exploiting potentials to improve sustainable water use in agriculture (Jägermeyr et al. 2017) may thus be highly important for maintaining the long-term stability of the global crop production system. It should also be noted that there is substantial potential to improve water use efficiency with integrated crop water management measures (Jägermeyr et al. 2016).

Interestingly, at the global level, increasing fertilizer use does not seem to decrease the sensitivity of crop yields to oscillations, potentially because low crop yield years remain the same while in years when conditions are suitable for crop growth, yields become even higher (Müller et al. 2018a), which would increase the sensitivity value as well. This explanation aligns with previous research, which has shown that increasing fertilizer use has limited potential to increase crop yields during years when weather conditions limit crop growth (Liebig's law). In other words, additional fertilizer use in years with unfavourable

seasonal climate condition does not lead to yield gain and is not cost effective, even if it is beneficial in normal conditions. Therefore, decision support systems which guide farmers about optimal fertilizer use under predicted growing season climate can be useful to avoid investments in fertilizers in bad years (Hayashi et al., 2018).

## 4.1 Limitations and way forward

The selection of the time windows for calculating the oscillation metric as well as defining the related growing seasons can have an impact on the spatial crop yield sensitivity footprint, as briefly illustrated in this study as well (Fig. 1, Fig. 11). Previously used approaches for identifying these relationships include e.g. looking at crop yield anomalies for the year (Heino et al. 2018, Iizumi et al. 2014, Yuan and Yamagata 2015) and years around (Anderson et al. 2017) strong oscillation anomalies. In these studies, strong oscillation anomalies are calculated either for the season in which the oscillations show their strongest

signal (Heino et al. 2018, Anderson et al. 2017, Yuan and Yamagata 2015) or the harvest season (Iizumi et al. 2014). In general, it can be said that it is very difficult to find metrics for the oscillations that would work perfectly everywhere. A lack of accurate, spatially-detailed crop calendars makes addressing this issue particularly challenging. The justification for the methods used here is to look at how crop yields vary around the time, in which these oscillations show their strongest signal, which can provide valuable information for early warning systems.

Future work could try to trace intermediate effects in order to explain the mechanisms at play, e.g. combining the effect of oscillations on weather, the effect of each aspect of weather on crop planting, development and harvest, and the final result in terms of crop yield. Such research could additionally provide useful information for decreasing crop yield variations, and thus increasing the resilience of crop production to climate variability.

The teleconnection patterns related to the IOD can be difficult to fully disentangle from ENSO due to their coevolution.

Previous studies have shown that around 20 % to 45 % of IOD variability could be explained by ENSO depending on the data and the investigated time frame (Saji and Yamagata 2003, Zhang et al. 2015). The nature of this relationship is still debated (Hameed et al. 2018, Stuecker et al. 2017), and determining the influence of ENSO on the IOD and vice-versa is not in the scope of this study. However, through the use of multivariate ridge regression, we aim to filter the influence of ENSO from the IOD patterns. Also, the relationship between ENSO and NAO has been studied, but that relationship has been shown to be

relatively weak (Hurrell et al. 2003).

The data used here are from state of the art global gridded crop models included in phase 1 of the GGCMI of AgMIP. However, major uncertainties in the simulated crop yields still exist, and the relationships observed here between crop yields and these oscillations are often not consistent throughout the ensemble of crop models (see Fig. 3). Differences and uncertainties among the models arise e.g. from soil and crop type parametrisations as well as handling of water and nutrient stress (e.g. Folberth et

al. 2019). Additionally, uncertainties in these GGCMs arise from the simulated cropping systems, as simulations assume only a single annual harvest per crop and per grid cell, whereas multiple harvests are common for e.g. rice. In general, simulated

crop yields seem to be most reliable in high nutrient-input areas (Müller et al. 2017), where observed climate variability also explains a majority of reported crop yield variation (Ray et al. 2015).

This study has included comparison with fully fertilized and irrigated management scenarios intended to capture (unattainable) ideal management, with no water or nutrient stress anywhere. This helps understand the physical potential of the management

measures for mitigating crop yield variability related to these oscillations, according to the models used. In future, practical limitations could also be taken into account by limiting water and fertilizer use to locally available resources.

The three climate oscillations included here are only a share of the whole range of periodically fluctuating climatological phenomena that could impact crop growing conditions. Thus, studying the relationship between simulated crop yields and other climate oscillations, not included here, such as Scandinavian Pattern or the Arctic Oscillation, would provide additional

insights to this topic, as demonstrated in a recent study by Ceglar et al. (2017).

## 5 Conclusions

This study strengthens the evidence that climate oscillations are drivers of crop yield variability around the world. In several areas, where these oscillations show robust impacts on crop production, e.g. Australia, southern Africa, as well as parts of North and South America, local risk reduction efforts as well as global efforts can already benefit from utilizing these known

relationships to improve the stakeholders' preparedness against crop production shocks associated with the climate oscillations. Information for maintaining the stability of global crop production is of high importance, given that anticipated climate change and population growth will keep increasing the pressure towards the global food system. Finally, our results suggest that increases (decreases) in the extent of irrigated area would, on average, reduce (amplify) the impacts of these oscillations on crop yields, which highlights the importance of sustainable water use in maintaining the long-term stability of the global crop

production system.

**Code and data availability**

The processing scripts are available from GitHub: https://github.com/matheino/crops_and_oscillations. The simulated crop yield data were retrieved from the GGCMI data archive: http://www.rdcep.org/research-projects/ggcmi, and they are also available through the links provided in the references of Table 1.

**Author contribution**

M.H., J.H.A.G. and M.K. designed the research in consultation with C.M. and T.I. Analyses were conducted by M.H. supported by all co-authors. M.H. wrote the article, with contributions from all co-authors.

## Competing interests

The authors declare no competing financial interests.

## Acknowledgements

We acknowledge the Agricultural Intercomparison and Improvement Project (AgMIP) for data provision. M.H. was financially supported by Maa- ja vesitekniikan tuki ry, the Vilho, Yrjö and Kalle Väisälä Foundation and AaltoENG doctoral programme, M.K. and J.H.A.G. were financially supported by Academy of Finland funded project WASCO (grant no. 305471), Emil Aaltonen Foundation funded project "eat-less-water" and M.K. additionally by Strategic Research Council (SRC) funded project 'From Failand to Winland' (grant no. 303623) as well as European Research Council (ERC) under the European Union's Horizon 2020 research and innovation programme (grant agreement No. 819202).

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
