# Peer review of "A multi-model analysis of teleconnected crop yield variability in a range of cropping systems"

_Earth System Dynamics, 2019_

## Referee Comment (RC1) · Anonymous Referee #1 · 9 Apr 2019

The authors use a suite of historical global gridded crop simulations from the AgMIP ensemble to examine the influence of natural climate oscillations on correlated crop yield impacts. Consistent with observed yield analyses, they find that ENSO variability can simultaneously affect nearly 50% of harvested areas for certain crops, while other modes of variability affect smaller areas but still have significant impacts. The authors suggest that this could help forecast climate shocks on the food system. Using additional simulations, they show that irrigation reduces the sensitivity to such climate variability but fertilizer application doesn't have a significant influence on reducing the climate sensitivity on these crops.

The study is an extension of work that has already been done by the authors on observed yields. While these simulations are helpful to isolate the role of climate variability and test scenarios of irrigation and fertilizer application, they do not provide a mechanistic explanation of the impacts or comparisons with the magnitude and extent of observed impacts. Although, I think this is a worthwhile study since crop models present some important tools to study the impacts of climate variability and management decisions, I have some major concerns about the methods and design of the study, which will affect the main conclusions. Before considering the merit of this manuscript for publication, I believe these following concerns need to be addressed.

Major Concerns:

- The three indices used include ENSO, IOD and NAO. All 4 crops studied here have almost identical sensitivities to ENSO and IOD, with some differences likely due to slightly different time periods used. I have a strong suspicion that this is because the ENSO and IOD indices have strong co-variability. Positive IOD's tend to develop during the development of positive ENSO phases (e.g Zhang et al., 2015, Stuecker et al. 2017). IOD's in the Fall are often followed by the peak of El Nino events in the winter. Based on how the time periods of the analyses are defined, this analysis is likely capturing the effect of El Ninos NOT IOD's.

- Relatedly, there needs to be some mechanistic explanation for how the NAO and IOD events influence yields in remote areas where they do not have strong (if any) climate teleconnections (reference Figure 1). For example, what is driving the yield sensitivity of crops in North America during IODs or in southern Africa during NAO events? I would recommend showing the underlying temperature and precipitation variability in response to each ENSO, IOD and NAO phases to support these findings.

- (Page 6, Section 2.3) In defining the oscillation specific harvest years, the evolution of the oscillations and their teleconnections in not correctly accounted for. The harvest years, in some places, cover multiple growing seasons. For example, IOD climate teleconnections do not typically last beyond the Fall season in which the IOD's occur into the following year's growing season, so including those subsequent seasons will provide spurious relationships. Similarly, El Nino's affect certain areas of the tropics such as South Asia strongly during the developing phase (Kumar et al., 2006). By defining the harvest year as starting on 1 December, these important connections are missed. Further, by extending them to the following growing season, when the impacts don't occur or as is stated in the manuscript, phase changes might occur, these sensitivities are likely to be spurious. For ENSO, it might make sense to define the harvest year from the growing season of the year it starts to develop to the start of the following year's growing season. For IOD and NAO, which are shorter lived, it might be more suitable to restrict harvest years to seasons when their impacts are known.

- (Page 6, Section 2.2) Regarding the El Nino index used, I would recommend using more commonly used metrics such as the Nino 3.4 index or at least test the sensitivity of your results to the Nino 3.4 Index, which is typically used to identify climate teleconnections.

- I realize that the models used here have been evaluated in a different paper. However, it would be useful to include an evaluation of the models in the supplement for metrics relevant here. For instance, how does each model capture observed yield various across global harvested areas. While the authors state that no model is obviously superior, they do not state whether any of them are capable of simulating observed yields.

- (Page 7, Section 2.4) The authors have used a linear regression model here. As far as I can tell, each mode is tested separately. However, given that they are related, I would think it would be more appropriate to have a multiple regression framework to isolate their individual influences.

- I am a bit flabbergasted at the inclusion of 24 maps in one figure (!!!). I would strongly recommend either splitting this plot by index or phase or crop. 24 is too many and I imagine others, like me, might have difficulty processing the information in Figure 2. Instead of a separate figure for agreement, it would be helpful to show agreement on

the maps in Figure 1 and 2, especially after splitting Figure 2. These changes will enhance the clarity of the figures and help decipher areas of model (dis)agreement more clearly.

- Page 14, Lines 15-20, It is a bit misleading to say that the sensitivity of crops to climate variability, increases with fertilizer application, given the discussion in these lines. If crop yields are improved during suitable climate conditions, that is a net positive, and it would be useful to have a metric to capture that improvement rather than suggest a negative effect of adding fertilizers.

- In Figure 4, the sensitivity of all crops is higher in the fully irrigated scenarios vs rainfed, based on Column 1. How does this suggest that irrigation reduces the sensitivity? This is likely just my confusion because of the way the information is presented in Figure 4. In Figure 4, is column 1, the difference in sensitivities of yield variability in the irrigation scenario – the rainfed scenario or vice versa? Does a positive difference suggest higher sensitivity in the irrigation scenario relative to the rainfed scenario?

- The conclusions will change if the analysis is changed to include the suggestions above. The discussion and conclusions sections will need to be edited accordingly.

———————————————

Minor Comments:

I encourage the author to include a discussion of the existing literature on the covari-ability of IOD/NAO and EL Nino indices.

Page 2 Line 16, what is the reference for IOD events being forecast months in advance?

Page 5 Line 1, what are the default model assumptions?

Page 5, Line 3, how are literature-based values different from default assumptions?

Page 7, Line 14, Where does the sample size number N=216 – 297 come from? Is that 12 models * number of simulation years somehow?

Section 2.5, What is the sample size for the comparison of the strong phases of each climate mode?

Page 10, first para, it would be useful to define the regions referred to in the discussion. For instance, I do not see wheat yield increases in "eastern South Asia" in the Fig. 1 as is suggested here.

Page 14, The result that irrigation reduces sensitivity of different crops makes sense. It would be helpful to have a metric that captures the relative areas of "actual" irrigation to explain the differences in sensitivity of different crops.

Figure 4, Please edit this figure for clarity. I would recommend either including boxes around each panel or lines to separate them. Also, please provide complete panel titles for the right 3 columns. Is this "actual - fully irrigated" scenarios?

In section 3, please refer to the specific panels in figure 4 in the discussion. I don't know which panel is being referred to the discussion, especially given the incomplete panel titles. I would also recommend doing this for other figures as much as possible.

––––––––––––––––––

References: Kumar, K. Krishna, B. Rajagopalan, M. Hoerling, G. Bates, M. Cane, 2006: Unraveling the Mystery of Indian Monsoon Failure During El Niño. Science, 314, 115-119.

Zhang, W., Wang, Y., Jin, F.‐F., Stuecker, M. F., and Turner, A. G. ( 2015), Impact of different El Niño types on the El Niño/IOD relationship, Geophys. Res. Lett., 42, 8570–8576, doi:10.1002/2015GL065703.

Stuecker, M. F., Timmermann, A., Jin, F.‐F., Chikamoto, Y., Zhang, W., Wittenberg, A. T., Widiasih, E., andZhao, S. ( 2017), Revisiting ENSO/Indian Ocean Dipole phase relationships, Geophys. Res. Lett., 44, 2481– 2492, doi:10.1002/2016GL072308.

––––––––––––––––––

---

## Referee Comment (RC2) · Daniel Kirk-Davidoff (Referee) · 1 Jul 2019

This manuscript describes a regression analysis that seeks to identify the global spatial response of modeled crop yields to global teleconnection pattern index variations. The ability of the authors' analysis to quantify the role of irrigation in damping the oscillations of crop yield due to climate variability seems potentially important. However, the results show some surprisingly strong responses in areas far afield from the action centers of some of the teleconnection patterns. For example, a strong response in the yield of maize to the Indian Ocean Dipole is observed along the US-Canadian border, while a strong response in the yield of maize and soybeans to the North America Oscillation is observed in Southeastern Australia. These are surprising, since I can't find any evidence of a significant relationship between the IOD and sensible weather in North

America, or between the NAO and sensible weather in Australia in global maps of these teleconnection patterns. It seems likely that these results are spurious, an accidental result of the large number of regions being modeled.

Before recommending this work for publication in Earth System Dynamics, I would like to see the a deeper exploration of the reliability of the relationships displayed. For example, it would be good see some scatterplots of the index values versus more directly relevant meteorological factors in each region (growing season length or precipitation) and of these factors versus yield, as well as between yield and index values, to get a sense of the predictive power of the relationships. Some other simple statistical tests would also be helpful. It would be good to see the whether the patterns of response of yield to teleconnection pattern presented in figures 1 and 2 are consistent when the timeseries are split into two parts (first half and second half). Finally, the authors should discuss at greater length the relative predictability of the various teleconnection patterns and how that convolves with level of uncertainty in the unlagged annual relationships presented here. If the NAO can only be predicted a few months in advance, what remaining skill is available for forecasting of the NAO's associated crop yield variability in advance of the harvest? It's one thing to note that if a strong NAO will be present, crop yields in some parts of the world will be a few percent above normal, but how much knowledge of crop yield anomalies is left if we only know that there's a 20% higher than normal chance of a strong NAO index averaged over next growing season?
* * *

---

## Author Comment (AC1) · 31 Aug 2019

We thank the editor and the reviewer for their careful evaluation of our manuscript and their constructive comments that helped us to improve the manuscript considerably. We have taken all the comments carefully into consideration when revising the paper. Please find our detailed responses to the review comments below. The main revisions include: 1) Assessing the sensitivity of crop yields to these oscillations by using a multivariate ridge regression framework, which controls for the co-variability of the oscillations; 2) Including an assessment about growing season weather teleconnections, and reflecting on how they relate to our main results; 3) Re-defining the allocation of annual growing seasons, so that the understanding of the teleconnections is better reflected in the analyses.

[Figure]

Responses to comments made by Reviewer #1:

R1.1.: The authors use a suite of historical global gridded crop simulations from the AgMIP ensemble to examine the influence of natural climate oscillations on correlated crop yield impacts. Consistent with observed yield analyses, they find that ENSO variability can simultaneously affect nearly 50% of harvested areas for certain crops, while other modes of variability affect smaller areas but still have significant impacts. The authors suggest that this could help forecast climate shocks on the food system. Using additional simulations, they show that irrigation reduces the sensitivity to such climate variability but fertilizer application doesn't have a significant influence on reducing the climate sensitivity on these crops.

The study is an extension of work that has already been done by the authors on observed yields. While these simulations are helpful to isolate the role of climate variability and test scenarios of irrigation and fertilizer application, they do not provide a mechanistic explanation of the impacts or comparisons with the magnitude and extent of observed impacts. Although, I think this is a worthwhile study since crop models present some important tools to study the impacts of climate variability and management decisions, I have some major concerns about the methods and design of the study, which will affect the main conclusions. Before considering the merit of this manuscript for publication, I believe these following concerns need to be addressed.

A1.1: Firstly, we want to thank the reviewer for the overall positive view of our study. We agree with the reviewer that using simulated crop yield data can provide important insights about climate impacts on crop productivity, while allowing to test for potential management options. However, we agree that the statistical method used does not allow mechanistic explanation of impacts. This is outside of our scope in this study.

As the reviewer notes, overall results are consistent with previous studies. Although, we relate our claims and results to existing knowledge throughout the text, detailed comparison with observed impacts is not performed – this is a highly complex task
given the uncertainties and confounding variables in the reported crop statistics (e.g. technological change, management decisions, pest outbreaks, multiple cropping, crop rotations) as well as modelled data. In the following sections we explain how we have addressed the issues raised by the reviewer.

R1.2.: The three indices used include ENSO, IOD and NAO. All 4 crops studied here have almost identical sensitivities to ENSO and IOD, with some differences likely due to slightly different time periods used. I have a strong suspicion that this is because the ENSO and IOD indices have strong co-variability. Positive IOD's tend to develop during the development of positive ENSO phases (e.g Zhang et al., 2015, Stuecker et al. 2017). IOD's in the Fall are often followed by the peak of El Nino events in the winter. Based on how the time periods of the analyses are defined, this analysis is likely capturing the effect of El Ninos NOT IOD's.

A1.2.: We want to thank the reviewer for pointing out that the results for IOD are potentially affected by its co-variability with ENSO. Therefore, instead of calculating the direct relationship between the IOD index and crop yields (as done in original submission), we now calculate the sensitivity of crop yield to IOD by controlling for ENSO as well as NAO by using multivariate ridge regression. The ridge regression framework was selected because it allows the explanatory variables (here oscillation indices) to correlate amongst each other.

After conducting the analysis with the updated method, for ENSO and NAO the patterns stayed relatively similar (Figure 1, some changes occurred also due to the changes made to the allocation of growing seasons), while for IOD the changes were larger, consistent with the reviewer's hypothesis that the previous IOD results were capturing the effect of ENSO. For the IOD some changes occurred e.g. in the Middle East and the Americas, however, the most important teleconnections (e.g. in Australia) still remained the same.

R1.3.: Relatedly, there needs to be some mechanistic explanation for how the NAO

and IOD events influence yields in remote areas where they do not have strong (if any) climate teleconnections (reference Figure 1). For example, what is driving the yield sensitivity of crops in North America during IODs or in southern Africa during NAO events? I would recommend showing the underlying temperature and precipitation variability in response to each ENSO, IOD and NAO phases to support these findings.

A1.3.: We agree with the reviewer that it is important to carefully reflect our results against existing understanding of the climate oscillations and their teleconnections. Hence, we have assessed how growing season temperature and soil moisture anomalies vary in relation to the oscillations included here (Figs S2-S3). In the main text, across Section 3.2, we then reflect on how the patterns found for weather variability relate to our main results.

R1.4.: (Page 6, Section 2.3) In defining the oscillation specific harvest years, the evolution of the oscillations and their teleconnections in not correctly accounted for. The harvest years, in some places, cover multiple growing seasons. For example, IOD climate teleconnections do not typically last beyond the Fall season in which the IOD's occur into the following year's growing season, so including those subsequent seasons will provide spurious relationships. Similarly, El Nino's affect certain areas of the tropics such as South Asia strongly during the developing phase (Kumar et al., 2006). By defining the harvest year as starting on 1 December, these important connections are missed. Further, by extending them to the following growing season, when the impacts don't occur or as is stated in the manuscript, phase changes might occur, these sensitivities are likely to be spurious. For ENSO, it might make sense to define the harvest year from the growing season of the year it starts to develop to the start of the following year's growing season. For IOD and NAO, which are shorter lived, it might be more suitable to restrict harvest years to seasons when their impacts are known.

A1.4.: In general, it is difficult to select the most appropriate harvest years for the oscillations, as different areas and crops have varying growing periods, which are impacted differently by the studied oscillations. For example, if the ENSO specific growing season was defined to begin at the beginning of the year, it could mask out some important maize and soybean teleconnections in northern South America, where, according to the models, maize and soybean planting occurs around January. Further, it should be noted that in GGCMI the models simulated only a single growing season annually.

The only way to convincingly resolve this issue is to fully unpick the mechanisms involved, including capturing temporal relationships between: 1) oscillation indices and weather, 2) climate and growing conditions, e.g. soil water availability, 3) growing conditions and yield, including variability of weather during the season potentially induced by the teleconnection. This is outside the scope of this paper.

However, we have re-defined the growing periods related to these oscillations so that the growing period for all these studied oscillations start in May and goes until the end of the following April, as this time period should capture the most important known teleconnections, e.g. in Australia, Asia and South America. It should be noted that due to these changes in the growing periods, some differences in the sensitivity direction occurred in the results for ENSO (Fig. 1), as the growing periods of e.g. soy and maize in the U.S. now fall in the period before ENSO would peak, while in the previous analyses their growing periods would've been after the ENSO peak.

R1.5.: Page 6, Section 2.2) Regarding the El Nino index used, I would recommend using more commonly used metrics such as the Nino 3.4 index or at least test the sensitivity of your results to the Nino 3.4 Index, which is typically used to identify climate teleconnections.

A1.5.: We have now conducted the same analysis using also the Niño 3.4 index (Figure S6), and the main results remain the same with this index as well. This helped increase the credibility of our results on ENSO impacts further. Thank you for the suggestion.

R1.6.: I realize that the models used here have been evaluated in a different paper. However, it would be useful to include an evaluation of the models in the supplement for metrics relevant here. For instance, how does each model capture observed yield various across global harvested areas. While the authors state that no model is obviously superior, they do not state whether any of them are capable of simulating observed yields.

A1.6.: We have now added a table (Table S1) adopted from Muller et al. (2017), which shows how well the simulated crop yields match reported yields from FAOSTAT at global level.

R1.7.: (Page 7, Section 2.4) The authors have used a linear regression model here. As far as I can tell, each mode is tested separately. However, given that they are related, I would think it would be more appropriate to have a multiple regression framework to isolate their individual influences.

A1.7.: Indeed, we agree that it is a good idea to use a multiple regression framework to account for the co-variability between the oscillations. Therefore, as described above (see reply to R1.2), we now use ridge regression to assess how the crop yields are impacted by these oscillations.

R1.8.: I am a bit flabbergasted at the inclusion of 24 maps in one figure (!!!). I would strongly recommend either splitting this plot by index or phase or crop. 24 is too many and I imagine others, like me, might have difficulty processing the information in Figure 2. Instead of a separate figure for agreement, it would be helpful to show agreement on the maps in Figure 1 and 2, especially after splitting Figure 2. These changes will enhance the clarity of the figures and help decipher areas of model (dis)agreement more clearly.

A1.8.: We appreciate that the number of maps can be overwhelming at first glance. The use of "small multiples" is, however, common in visualisation as an effective way of providing a quick overview so that large quantities of data can be compared. In this case, the figure allows comparison of spatial patterns across crops, phases and indices. Splitting by either of these would then hamper comparison. At the same time, the casual reader does not need to identify parameters by themselves – the

patterns are discussed in text. To clarify this point, the caption now adds: "Patterns are discussed in Sections 3.1 and 3.2."

Rather than modifying Figure 3, we have tried to clarify its role in the paper. The original caption indicated that it shows "agreement" between models and methods, which falsely gave the impression that it should be primarily read with Figure 1 and 2. On the contrary, it is intended as a robust summary of locations where indices and yields are related. The caption now reads: "Summary of relationship between ENSO, IOD, and NAO and crop yields across models and methods".

R1.9.: Page 14, Lines 15-20, It is a bit misleading to say that the sensitivity of crops to climate variability, increases with fertilizer application, given the discussion in these lines. If crop yields are improved during suitable climate conditions, that is a net positive, and it would be useful to have a metric to capture that improvement rather than suggest a negative effect of adding fertilizers.

A1.9.: We apologize that we have not clearly communicated our results. Indeed, we don't imply that fertilizers are not useful in increasing crop yields, but merely that they may not be effective in mitigating weather related impacts; i.e. we were only discussing potential reasons, why these the numbers come out this way. We now mention this in the main text: "Note that this does not mean fertiliser fails to improve crop yields – only that it does not lead to more stable yields in the face of weather variability" (Page 14, Lines 23-24).

R1.10.: In Figure 4, the sensitivity of all crops is higher in the fully irrigated scenarios vs rainfed, based on Column 1. How does this suggest that irrigation reduces the sensitivity? This is likely just my confusion because of the way the information is presented in Figure 4. In Figure 4, is column 1, the difference in sensitivities of yield variability in the irrigation scenario – the rainfed scenario or vice versa? Does a positive difference suggest higher sensitivity in the irrigation scenario relative to the rainfed scenario?

A1.10.: We have now clarified the figure so that the panel titles indicate that the sensitivity differences are calculated as change in sensitivity relative to a baseline, which is the Fully irrigated scenario for the 1st panel, and the Actual scenario for the four other panels.

R1.11.: The conclusions will change if the analysis is changed to include the suggestions above. The discussion and conclusions sections will need to be edited accordingly.

A1.11.: We have modified the Discussion and Conclusions sections to reflect our revised results. However, as our main results did not change significantly, no large changes were needed in these sections.

R1.12.: I encourage the author to include a discussion of the existing literature on the covariability of IOD/NAO and EL Nino indices.

A1.12.: We have now included a paragraph discussing the covariability between IOD (as well as briefly NAO) and ENSO in the Discussion.

R1.13.: Page 2 Line 16, what is the reference for IOD events being forecast months in advance?

A1.13.: Two studies by Luo et al. (2005 & 2008) show successful prediction of three IOD events (1994, 2006 and 2007) with seasonal lead time. However, as these results don't show that the status of IOD can be predicted at all times, the sentence was rephrased: "As the phase and development of ENSO, IOD and NAO can potentially be forecasted from several months (IOD, NAO (Luo et al. 2008, Scaife et al. 2014)) up to one year (ENSO (Luo et al., 2005, Ludescher et al. 2014)) in advance, –" (Page 2, Lines 17-18).

R1.14.: Page 5 Line 1, what are the default model assumptions?

A1.14.: The default model configurations are based on the management and technology assumptions typically used by the modelling groups for historical simulations. Hence, for the default configuration the GGCMI coordinators allowed the modelling

teams to define their own assumptions for setting up the models. This is now more explicitly mentioned in the text explained: "To account for varying assumptions of growing season and fertilizer use, in GGCMI, model simulations were conducted for three configurations: standard model assumptions (default), harmonized growing season and nutrient input (fullharm), and harmonized growing season with no nutrient limitation (harm-suffN). For the default configuration each modelling group were instructed to use their own model assumptions" (Page 4 Line 6 - Page 5 Line 3).

R1.15.: Page 5, Line 3, how are literature-based values different from default assumptions?

A1.15.: For the harmonized configurations the growing season and assumptions about fertilizer use are fixed (based on reported patterns from literature) among the models, while for the default configuration the modelling teams were allowed to use their own normal assumptions, which can be dynamic (e.g. planting dates can change depending on pre-season weather condition).

R1.16.: Page 7, Line 14, Where does the sample size number N=216 – 297 come from? Is that 12 models * number of simulation years somehow?

A1.16.: This is correct. The sample used for the regression is such that the crop yield time series of all the models is used for fitting the regression. This is now been more explicitly explained: "For the main analysis (actual scenario)), the regression was calculated for each FPU separately using crop yield anomaly time series from all GGCMs that simulate the crop in question with the AgMERRA climate input (N=216-297, depending on crop). Hence, we utilize the crop yield time series of all the models in fitting the regression." (Page 7, Lines 15-17).

R1.17.: Section 2.5, What is the sample size for the comparison of the strong phases of each climate mode?

A1.17.: The strong negative and positive phases were defined based on the 25% and

75% percentiles of the indices. Thus, sample sizes were 54-74, depending on the crop in question. This is now explicitly mentioned in the revised text: "Strongly negative (positive) phases of the oscillations were defined as the years when the respective oscillation index was smaller (larger) than the 25th (75th) percentile of all yearly index values (Nanomaly = 54-74, depending on crop)" (Page 8, Lines 8-10).

R1.18.: Page 10, first para, it would be useful to define the regions referred to in the discussion. For instance, I do not see wheat yield increases in "eastern South Asia" in the Fig. 1 as is suggested here.

A1.18.: We have now added a map about the regions we refer to in the text (Figure S1).

R1.19.: Page 14, The result that irrigation reduces sensitivity of different crops makes sense. It would be helpful to have a metric that captures the relative areas of "actual" irrigation to explain the differences in sensitivity of different crops.

A1.19.: The aim of this analysis was to look into the potential for changes in agricultural inputs, from the current baseline, to change the sensitivity of crop yields to these oscillations. Therefore, the current management conditions do impact the numbers obtained. We have included information about the current extent of irrigated areas to inform the reader that they do indeed affect the results: "This ranking is expected, as the majority of rice harvested areas are irrigated (62% globally) and soybean has the smallest irrigated area share of these four crops (8%), while maize (21%) and wheat (31%) fall in between (Portmann et al. 2010)" (Page 14, Lines 8-10).

R1.20.: Figure 4, Please edit this figure for clarity. I would recommend either including boxes around each panel or lines to separate them. Also, please provide complete panel titles for the right 3 columns. Is this "actual - fully irrigated" scenarios?

A1.20.: We apologize that the figure was not clearly set up, and thank for the suggestions made to improve the figure. We have now divided the panels using shading,

included the titles for all the panels, as well as updated the panel titles to better reflect the analyses conducted.

R1.21.: In section 3, please refer to the specific panels in figure 4 in the discussion. I don't know which panel is being referred to the discussion, especially given the incomplete panel titles. I would also recommend doing this for other figures as much as possible.

A1.21.: We have now included more specific references to Figure 4, when discussing its results.

R1.22.: References:

Kumar, K. Krishna, B. Rajagopalan, M. Hoerling, G. Bates, M. Cane, 2006: Unraveling the Mystery of Indian Monsoon Failure During El Niño. Science, 314, 115-119. Zhang, W., Wang, Y., Jin, F.âËŸARËĞ F., Stuecker, M. F., and Turner, A. G. ( 2015), Impact of different El Niño types on the El Niño/IOD relationship, Geophys. Res. Lett., 42, 8570–8576, doi:10.1002/2015GL065703.

Stuecker, M. F., Timmermann, A., Jin, F.âËŸARËĞ F., Chikamoto, Y., Zhang, W., Wittenberg, A. T., Widiasih, E., andZhao, S. ( 2017), Revisiting ENSO/Indian Ocean Dipole phase relationships, Geophys. Res. Lett., 44, 2481– 2492, doi:10.1002/2016GL072308.

A1.22.: Thank you for directing us to these references. The references used in our replies, are listed below:

Müller, C., Elliott, J., Chryssanthacopoulos, J., Arneth, A., Balkovic, J., Ciais, P., Deryng, D., Folberth, C., Glotter, M., Hoek, S., Iizumi, T., Izaurralde, R. C., Jones, C., Khabarov, N., Lawrence, P., Liu, W., Olin, S., Pugh, T. A. M., Ray, D. K., Reddy, A., Rosenzweig, C., Ruane, A. C., Sakurai, G., Schmid, E., Skalsky, R., Song, C. X., Wang, X., De Wit, A. and Yang, H.: Global gridded crop model evaluation: Benchmarking, skills, deficiencies and implications, Geoscientific Model Dev., 10, 1403-1422,

[Figure]

10.5194/gmd-10-1403-2017, 2017.

Luo, J., Masson, S., Behera, S., Shingu, S. and Yamagata T.; Seasonal climate predictability in a coupled OAGCM using a different approach for ensemble forecasts, J. Climate, 18, 4474–4497, 2005. Luo, J., Behera, S., Masumoto, Y., Sakuma, H. and Yamagata, T.: Successful prediction of the consecutive IOD in 2006 and 2007, Geophys. Res. Lett., 35, 2008.

---

## Author Comment (AC2) · 31 Aug 2019

We thank the editor and the reviewer for their careful evaluation of our manuscript and their constructive comments that helped us to improve the manuscript considerably. We have taken all the comments carefully into consideration when revising the paper. Please find our detailed responses to the review comments below. The main revisions include: 1) Assessing the sensitivity of crop yields to these oscillations by using a multivariate ridge regression framework, which controls for the co-variability of the oscillations; 2) Including an assessment about growing season weather teleconnections, and reflecting on how they relate to our main results; 3) Re-defining the allocation of annual growing seasons, so that the understanding of the teleconnections is better reflected in the analyses.

[Figure]

Responses to comments made by the editor:

R2.1.: This manuscript describes a regression analysis that seeks to identify the global spatial response of modeled crop yields to global teleconnection pattern index variations. The ability of the authors' analysis to quantify the role of irrigation in damping the oscillations of crop yield due to climate variability seems potentially important.

A2.1.: Firstly, we want to thank the editor for the overall positive view of our study. We agree with the editor that the data used in this study provides important insights into the role of irrigation in damping climate impacts on crop productivity.

R2.2.: However, the results show some surprisingly strong responses in areas far afield from the action centers of some of the teleconnection patterns. For example, a strong response in the yield of maize to the Indian Ocean Dipole is observed along the US-Canadian border, while a strong response in the yield of maize and soybeans to the North America Oscillation is observed in Southeastern Australia. These are surprising, since I can't find any evidence of a significant relationship between the IOD and sensible weather in North America, or between the NAO and sensible weather in Australia in global maps of these teleconnection patterns. It seems likely that these results are spurious, an accidental result of the large number of regions being modeled.

A2.2.: The editor is correct that due to the large number of areas being modelled, some false positives for our statistical tests are expected. However, we don't make any conclusions or recommendations based on our analysis alone, but reflect on how our results relate to the current knowledge before drawing conclusions.

Also, in addition to analyzing how crop yields vary with these oscillations, we have now included analyses about the sensitivity (using multivariate ridge regression) of temperature and soil moisture anomalies to these oscillations (Figure S2-S3). Based on these results, there seems to be a statistical relationship between IOD and weather in North America (see also Fig. 21 in Saji and Yamagata 2003) as well as between NAO and temperature conditions in Australia.
R2.3.: Before recommending this work for publication in Earth System Dynamics, I would like to see the a deeper exploration of the reliability of the relationships displayed. For example, it would be good see some scatterplots of the index values versus more directly relevant meteorological factors in each region (growing season length or precipitation) and of these factors versus yield, as well as between yield and index values, to get a sense of the predictive power of the relationships.

A2.3.: We agree with the editor that it is important to analyse the reliability of our results, and therefore already in the original submission, we included a relatively thorough analysis about the uncertainty of our results related to the gridded crop model ensemble used here (especially Figure 3).

As described above, we have now included an assessment about how soil moisture and temperature variability is related to these oscillations. Further, as the analysis includes over 500 spatial units, instead of providing scatter plots about the relationships, we provide the R2 values for the regression results (Figure S13), which show that e.g. in Australia a substantial proportion of crop yield variability can be explained with the oscillations studied here. More extensive exploration of relationships with directly relevant meteorological factors is out of scope, as it risks giving the reader the impression that we understand the mechanisms involved better than we actually do.

R2.4.: Some other simple statistical tests would also be helpful. It would be good to see the whether the patterns of response of yield to teleconnection pattern presented in figures 1 and 2 are consistent when the timeseries are split into two parts (first half and second half).

A2.4.: We want to thank the editor for the suggestion. However, the statistical significance of the sensitivity values is already assessed by bootstrapping, which means that, for each spatial unit, we have calculated the regression for 1000 sub-samples of the crop yield data (explained in Page 7, Lines 20-22). This is a more thorough alternative to split-sample testing, and we therefore expect to find statistically significant sensitivity

values in the same areas even if the time series is split in half.

R2.5.: Finally, the authors should discuss at greater length the relative predictability of the various teleconnection patterns and how that convolves with level of uncertainty in the unlagged annual relationships presented here. If the NAO can only be predicted a few months in advance, what remaining skill is available for forecasting of the NAO's associated crop yield variability in advance of the harvest? It's one thing to note that if a strong NAO will be present, crop yields in some parts of the world will be a few percent above normal, but how much knowledge of crop yield anomalies is left if we only know that there's a 20% higher than normal chance of a strong NAO index averaged over next growing season?

A2.5.: We fully agree that there is a long way to go before reliable forecasting is possible. This study only provides background knowledge on the (possible) existence of relationships. We have added a paragraph discussing the usefulness and limitations of our results in mitigating climate impacts on crop yields and society. In the paragraph we e.g. state that: "In Australia, there is significant potential to utilize the information of IOD along with ENSO, to understand crop yield fluctuations, as they can explain a large proportion of local crop yield variability (Fig. S13, Yuan and Yamagata 2015). – However, the quality of predictions of this type would naturally depend on the skill of the climate forecasts as well as the strength of the teleconnection. This study only provides a first assessment of correlations, and further work is needed before reliable forecasts can be provided. "(page 16, Lines 10-16).

References:

Saji, N. H. and Yamagata, T.: Possible impacts of Indian Ocean dipole mode events on global climate, Climate Research, 25, 151-169, 2003.

Yuan, C. and Yamagata, T.: Impacts of IOD, ENSO and ENSO Modoki on the Australian winter wheat yields in recent decades, Scientific reports, 5, 2015.

---

## Author Response (AR2)

Authors' responses on reviewers' comments

**esd-2019-8: A multi-model analysis of teleconnected crop yield variability in a range of cropping systems**

Matias Heino, Joseph H. A. Guillaume, Christoph Müller, Toshichika Iizumi, and Matti Kummu

We are delighted to hear that the editor and the reviewer were satisfied with our replies and revisions. We want to thank the reviewer and the editor for their comments and suggestions, as they have improved the quality of this study considerably. Please, find our responses to the reviewers' remarks below.

**R1.1** The data used in the regression is a bit unclear to me. Since the model only simulates one growing season, this means that there is only one harvest per year. Are the climate indices used in the regression based on the harvest year or the sowing year, when they are different? Also, if there is only one growing season per year, why do the results change if the growing season is defined starting in May versus in January?

**A1.1** Usually, an annual crop yield value is allocated to the year that the crop is harvested (Müller et al. 2017). Here, to align with the timings of these climate oscillations, that affect crop growth during the growing season not only at the moment of harvest, we have allocated the crop yield values based on sowing dates, where the "harvest year $t$" refers to the period from the beginning of May (year $t$) until the end of April (year $t+1$). For ENSO and NAO, the indices are calculated for December (year $t$), January (year $t+1$), and February (year $t+1$), while for the IOD, the index is calculated for September, October, and November (year $t$).

The results can change depending on how the crop yield values are temporally allocated. For example, areas where crops are planted after the beginning of May (year $t$), and harvested before December (year $t$), crop yields are now allocated to year $t$, but were previously allocated to year $t-1$. This would be the case for example for maize in some parts of the United States, where, according to the model input data, maize is planted after May, and harvested before December. As for example a La Niña event is often preceded by an El Niño event (Anderson et al. 2017), this could have an effect on the observed sensitivity direction as well.

We have revised the manuscript to clarify and discuss the points mentioned above (Page 6 Lines 10-11 and Line 20 and Page 18 Lines 5-14).

**R1.2.** Please provide some discussion of how the impacts of the same index on different crops can vary in the same region. For instance, in Line 20-22 on page 12, it is stated that wheat and maize have negative sensitivity to NAO but Soybean has the opposite. Is that because these are grown at different times of the year or because these crops respond differently to the NAO induced climatic conditions? Presumably negative soil moisture would be bad for all three crops.

**A1.2** It is empirically known that the same climate oscillation index can impact distinct crops differently (Iizumi et al., 2014). However, revealing the exact reasons are challenging because of many potentially contributing factors. For soybean in the Middle East, based on the model input data, sowing dates vary from location to location (in some areas soybean sowing occurs in Spring before May, while in other areas soybean is planted later in the year) and also depending on the irrigation set-up used. This instability in sowing dates might have an impact on the observed signal,

compared to maize and wheat which have spatially more stable growing seasons. We have now elaborated on these points in the manuscript (Page 11, Lines 15-19).

All in all, the question here relates to a fundamental issue regarding this study (raised in the original manuscript and discussed during the review process as well). The ultimate aim of research on this topic is to provide a mechanistic understanding of the effect of climate oscillations on crop yield, mediated by weather throughout the growing season. This is, however, too complex a task at this time, and this study is therefore limited to providing correlation-based evidence of association between climate oscillations and crop yield (with additional analyses of association of climate oscillations and weather added at the reviewers' request). Substantial assumptions are therefore made regarding treatment of temporal relationships (comment R1.1), and the ability to explain results is limited (comment R1.2). We trust that with the additional changes mentioned, the nature of this work is now sufficiently clear to the reader.

**References**

Anderson, W., Seager, R., Baethgen, W. and Cane, M.: Crop production variability in North and South America forced by life-cycles of the El Niño Southern Oscillation, Agric. For. Meteorol., 239, 151-165, 2017.

Iizumi, T., Luo, J., Challinor, A. J., Sakurai, G., Yokozawa, M., Sakuma, H., Brown, M. E. and Yamagata, T.: Impacts of El Niño Southern Oscillation on the global yields of major crops, Nature communications, 5, 3712, 2014.

Müller, C., Elliott, J., Chryssanthacopoulos, J., Arneth, A., Balkovic, J., Ciais, P., Deryng, D., Folberth, C., Glotter, M., Hoek, S., Iizumi, T., Izaurralde, R. C., Jones, C., Khabarov, N., Lawrence, P., Liu, W., Olin, S., Pugh, T. A. M., Ray, D. K., Reddy, A., Rosenzweig, C., Ruane, A. C., Sakurai, G., Schmid, E., Skalsky, R., Song, C. X., Wang, X., De Wit, A. and Yang, H.: Global gridded crop model evaluation: Benchmarking, skills, deficiencies and implications, Geoscientific Model Dev., 10, 1403-1422, 10.5194/gmd-10-1403-2017, 2017.

Sacks, W. J., Deryng, D., Foley, J. A., & Ramankutty, N.: Crop planting dates: an analysis of global patterns. Global Ecology and Biogeography, 19(5), 607-620. 2010.